# Bacteriophage Rescue Therapy of a Vancomycin-Resistant *Enterococcus faecium* Infection in a One-Year-Old Child following a Third Liver Transplantation

**DOI:** 10.3390/v13091785

**Published:** 2021-09-07

**Authors:** Kevin Paul, Maya Merabishvili, Ronen Hazan, Martin Christner, Uta Herden, Daniel Gelman, Leron Khalifa, Ortal Yerushalmy, Shunit Coppenhagen-Glazer, Theresa Harbauer, Sebastian Schulz-Jürgensen, Holger Rohde, Lutz Fischer, Saima Aslam, Christine Rohde, Ran Nir-Paz, Jean-Paul Pirnay, Dominique Singer, Ania Carolina Muntau

**Affiliations:** 1Department of Pediatrics, Kinder-UKE, University Medical Center Hamburg-Eppendorf, 20246 Hamburg, Germany; t.harbauer@uke.de (T.H.); s.schulz-juergensen@uke.de (S.S.-J.); dsinger@uke.de (D.S.); muntau@uke.de (A.C.M.); 2Burn Centre, Laboratory for Molecular and Cellular Technology (LabMCT), Queen Astrid Military Hospital, B-1120 Brussels, Belgium; maia.merabishvili@mil.be; 3Institute of Dental Sciences, School of Dentistry, Hebrew University of Jerusalem, Jerusalem 9112001, Israel; dnlgelman@gmail.com (D.G.); leronK@ekmd.huji.ac.il (L.K.); ortal.yerushalmy@mail.huji.ac.il (O.Y.); shunitc@ekmd.huji.ac.il (S.C.-G.); 4Institute of Medical Microbiology, Virology and Hygiene, University Medical Center Hamburg-Eppendorf, 20246 Hamburg, Germany; mchristner@uke.de (M.C.); rohde@uke.de (H.R.); 5Department of Visceral Transplantation, University Medical Center Hamburg-Eppendorf, 20246 Hamburg, Germany; u.herden@uke.de (U.H.); l.fischer@uke.de (L.F.); 6Center for Innovative Phage Applications and Therapeutics, Division of Infectious Diseases and Global Public Health, University of California, San Diego, CA 92093, USA; saslam@health.ucsd.edu; 7Leibniz Institute DSMZ—German Collection of Microorganisms and Cell Cultures GmbH, 38124 Braunschweig, Germany; chr@dsmz.de; 8Department of Clinical Microbiology and Infectious Disease, Hadassah University Medical Center, Faculty of Medicine, Hebrew University of Jerusalem, Jerusalem 9112001, Israel; nirpaz@hadassah.org.il

**Keywords:** bacteriophage, *Enterococcus faecium*, biliary atresia, vancomycin, multi-drug resistance, critical care, liver transplantation, pediatric

## Abstract

Phage therapy is an experimental therapeutic approach used to target multidrug-resistant bacterial infections. A lack of reliable data with regard to its efficacy and regulatory hurdles hinders a broad application. Here we report, for the first time, a case of vancomycin-resistant *Enterococcus faecium* abdominal infection in a one-year-old, critically ill, and three times liver transplanted girl, which was successfully treated with intravenous injections (twice per day for 20 days) of a magistral preparation containing two *Enterococcus* phages. This correlated with a reduction in baseline C-reactive protein (CRP), successful weaning from mechanical ventilation and without associated clinical adverse events. Prior to clinical use, phage genome was sequenced to confirm the absence of genetic determinants conferring lysogeny, virulence or antibiotic resistance, and thus their safety. Using a phage neutralization assay, no neutralizing anti-phage antibodies in the patient’s serum could be detected. Vancomycin-susceptible *E. faecium* isolates were identified in close relation to phage therapy and, by using whole-genome sequencing, it was demonstrated that vancomycin-susceptible *E. faecium* emerged from vancomycin-resistant progenitors. Covering a one year follow up, we provide further evidence for the feasibility of bacteriophage therapy that can serve as a basis for urgently needed controlled clinical trials.

## 1. Introduction

Biliary atresia is an obliterative cholangiopathy presenting in the neonatal period, and it is treated by Kasai portoenterostomy during the first months of life as first-line therapy [1]. However, progressive cholestasis and recurrent cholangitis, even after portoenterostomy, make biliary atresia the leading indication for liver transplantation in the pediatric cohort [2]. Abdominal site infections caused by enterococci belong to the most common complications after Kasai portoenterostomy [3] and liver transplantation [4,5]. The clinical management of infections with *E. faecium* is particularly challenging due to the high rate of multidrug resistance in this species [4,5,6]. In general, vancomycin-resistant enterococci are a major medical concern, with *E. faecium* being one of the ESKAPE pathogens classified as “high priority” in the WHO priority pathogens list [7].

Facing emerging rates of infections by multidrug-resistant (MDR) bacteria, phage therapy has gained worldwide attention to serve as a possible treatment option. The growing worldwide interest in phage therapy, which was predominantly used in Eastern Europe during the past century [8], is illustrated by recent reports of cases or case series of successful phage therapy in refractory infections, e.g., [9,10,11,12]. However, to our knowledge, to date no intravenous phage therapy of enterococcal infection has been reported. Although phage therapy is increasingly seen as a promising alternative or complement to antibiotic therapy, its efficacy has not been shown in controlled clinical trials. This results in regulatory hurdles, which contribute to limit a broader application of phage therapy [13,14]. Here we describe the successful phage therapy of a vancomycin-resistant *E. faecium* (VRE*fm*) infection in a pediatric liver transplant recipient, underscoring the potential usefulness of phage therapy in the treatment of infections caused by VRE.

## 2. Materials and Methods

### 2.1. Phage Susceptibility Testing

Phage susceptibility testing was performed in triplicate using a modified double agar overlay method [15]. Briefly, 0.1 mL of an overnight culture of the target bacteria VRE*fm* was added to 3.5 mL of agarose (0.5%) and plated on brain heart infusion (BHI, Difco, Detroit, MI) agar plates. Drops (0.05 mL) of 27 anti-enterococci phages from the Israeli Phage Bank (IBP) [16] in titers of ~10^8^–10^9^ plaque-forming units (PFU)/mL were spotted on the bacterial lawn. The plates were incubated for 18 h at 37 °C. Two phages, EFgrKN (GenBank Accession: MW004544) and EFgrNG (GenBank Accession: MW004545) [17], exerted clear plaques, and their efficacy was validated in liquid culture (Appendix A, and also as schematical figure in [18]). Next, the efficacy of the phages was tested in triplicate in the presence of antibiotics to determine synergy or interference between them. To this end, untreated and treated bacteria were grown in a 96-well plate at 37 °C for 24 h. The 600 nm absorbance was recorded every 20 min after 5 s linear shaking. The bacteria were added in their logarithmic phase (1.5 × 10^7^ colony forming units (CFU)/mL). Phages and antibiotics were added at t = 0 as follows: EFGrNG: 7 × 10^7^ PFU/mL; EFGrKN: 2.4 × 10^8^ PFU/mL; ampicillin: 16 µg/mL (0.5 MIC); vancomycin: 16 µg/mL (0.5 MIC); chloramphenicol: 1 µg/mL (~0.5 MIC of linezolid); gentamicin: 12.5 µg/mL (~0.025 MIC because the bacteria were highly sensitive).

### 2.2. Phage DNA Analysis and Lysogeny

DNA of the phages was extracted, purified and sequenced as previously described [19]. Analysis of lysogeny, virulence and antibiotic resistance determinants was performed using Abricate (version 0.8.13, [Seemann T. Abricate Github https://github.com/tseemann/abricate (accessed on 30 July 2021)]), comparing to all of its databases.

### 2.3. Production of Phage Active Pharmaceutical Ingredients (APIs)

Phages EFgrKN and EFgrNG were propagated by the double agar overlay method [15] to a titer of 10^10^ PFU/mL using bacterial host strain VRE*fm* isolated from the patient. The obtained lysates were centrifuged at 35,000× *g* for 1.5 h, and the phage pellet was resuspended in DPBS (Lonza, Verviers, Belgium) to obtain phage stocks with a titer of 10^11^ PFU/mL, which were further diluted to 10^9^ PFU/mL and endotoxin purified by Endotrap HD (Lionex, Braunschweig, Germany) column mode affinity chromatography. Samples of the obtained phage Active Pharmaceutical Ingredients (APIs) were sent to Sciensano, the Belgian Scientific Institute of Public Health (Brussels, Belgium), for quality assessment and product certification [20]. Upon Sciensano approval, both phage APIs were mixed and diluted, in the form of a magistral preparation, to the titers of ~10^7^ and ~10^8^ PFU/mL in 0.9 % NaCl (Fresenius Kabi, Bad Homburg, Germany) and sent directly to the University Medical Center Hamburg—Eppendorf for application in the patient. As the two phages produce indistinguishable plaque morphologies, it was impossible to determine the titer of each phage separately when present in the mixture. The joint titers of the first and the second batches of magistral preparations were defined as 8.1 × 10^7^ and 5.2 × 10^8^ PFU/mL, respectively.

### 2.4. Stability of the Phages

The stability of the two batches of magistral preparations containing phages EFgrKN and EFgrNG was determined by the double agar overlay method [15]. The preparations were stored in 15 mL polypropylene tubes (Greiner Bio-One, Vilvoorde, Belgium) at 4 °C. As noted earlier, only joint titers were defined.

### 2.5. Phage Neutralization Assay

Phage neutralization by the patient’s serum was evaluated according to Adams 1959 [21] with some modifications. Blood samples were collected on 4, 28 and 49 days after initiation of phage therapy and were centrifuged, after clotting, at 2000× *g* for 10 min. The obtained serum samples (supernatant) were stored at −80 °C. For testing, 0.9 mL of the diluted (1:100) serum samples was mixed with 0.1 mL of phages EFgrKN or EFgrNG at the concentrations of 4.6 ± 2.4 × 10^7^ and 4.3 ± 2.7 × 10^7^ PFU/mL, respectively, and incubated at 37 °C for 30 min. After incubation, the phages were titered using the host strain VRE*fm*, which was isolated from the patient, to determine the number of non-neutralized active phage particles. Each sample was tested in triplicate against each phage, and mean values and standard deviations were determined.

The rate of phage inactivation is calculated using the following equation: K = 2.3 D/t × log p0/p, in which D is the reciprocal of serum dilution, p0 the initial number of phages and p the final number of phages at time t min. The equation is only valid when the neutralization rate of the phage is within the range 90–99%.

### 2.6. Microbiological Methods and Whole Genome Sequencing of E. faecium Isolates

Culture, species identification and susceptibility testing were carried out as previously described [22]. For whole-genome analysis, Nextera XT libraries were sequenced on an Illumina NextSeq 500 platform to obtain paired 150 bp reads with at least 100-fold coverage. De novo assembly, annotation, MLST-typing, resistome analysis and core-genome snip distance calculation were performed with the nullarbor toolchain (version 2.0.20191013, [Seemann T, Goncalves da Silva A, Bulach DM, Schultz MB, Kwong JC, Howden BP. Nullarbor Github https://github.com/tseemann/nullarbor]) (accessed on 30 July 2021). Large-scale genome comparisons were performed with mauve (version 2.4.0) [23].

## 3. Results

### 3.1. Case Presentation

The patient first presented to University Medical Center Hamburg—Eppendorf in 2019 at the age of 10 months and had undergone portoenterostomy due to biliary atresia at the age of 8 weeks in Iran.

At that time, her clinical picture was compatible with that of a failed Kasai, showing a severely cirrhotic liver and additional multiple liver abscesses after recurrent cholangitis. She was immediately evaluated for liver transplantation and, due to ongoing deterioration, subsequently received a left lateral liver split five weeks after admission as a high-urgency transplantation. For long-term immunosuppression, cyclosporin A (CSA) and prednisolone were used. Vast intrahepatic bacterial colonization led to a severe systemic infection post transplantation, which was, among other transiently detected bacteria, mainly caused by vancomycin-resistant *E. faecium* (VRE*fm*), grown from various specimens (i.e., explanted organ, blood cultures, abdominal drainage, bile ducts of the transplanted organ). With extended life support and adaption of antibiotics (Figure 1), stabilization of her vital functions and resolution of the systemic infection was achieved.

However, even repetitive abdominal lavages and revision of the biliodigestive anastomosis did not resolve the VRE*fm*-related abdominal site infection. Liver ischemia reperfusion injury with delayed graft function, together with a critical microvascular oxygen supply during the post-transplant period, resulted in superinfected necrotic areas.

Due to progressive organ damage, and to eradicate the reservoir of the infection, a re-transplantation was needed five weeks after the first transplantation, which was performed with a full organ. Unfortunately, after a temporary improvement in liver function and systemic inflammation, VRE*fm* was repeatedly detected in abdominal swabs, and the patient’s medical condition deteriorated to a level similar to that before the re-transplantation, as demonstrated by blood CRP and bilirubin levels. Multiple antibiotic treatment schemes (Figure 1) and extensive surgical treatment were not sufficient to control the infection. Based on recent reports of successful treatments of refractory infections with bacteriophages in transplant patients [9,11] we explored this option for our patient. Meanwhile, because of terminal transplant organ damage in a mildly ventilated child, without need for vasopressors and a localized infection of the abdomen, listing of the patient for a third liver transplantation was decided after multiple interdisciplinary case conferences and thorough informed consent from the parents. The third transplantation with a split organ was performed one month after the second transplantation together with a splenectomy, since previous imaging studies were highly suggestive for multiple intra-splenic abscesses. Indeed, post-operative bacterial cultures confirmed a severe splenic colonization with VRE*fm*.

The third transplantation plus splenectomy led to a persistent normalization of liver function and cholestasis, but VRE*fm* was still detected in the abdomen, with ongoing elevated CRP levels.

Three weeks after the third liver transplantation (Figure 1 day 110), we applied an individualized two-phage cocktail prepared for use as salvage therapy. The indication of phage therapy was accompanied and approved by the Ethics Committee of the University Medical Center Hamburg–Eppendorf, and it was conducted under the umbrella of article § 37 (Unproven Interventions in Clinical Practice) of the Declaration of Helsinki after expert advice and informed consent of the family. We initiated a ten-day course of 2 mL/kg bodyweight (BW) of the magistral preparation (joint titer 8.1 × 10^7^ PFU/mL in NaCl 0.9%) administered intravenously over two hours, twice daily. To reduce the theoretical risk of an allergic reaction against the phage preparation, an H_1_-antagonist was administered before every phage application. In addition, the first phage dose was given over a duration of 4 h and after prior administration of methylprednisolone (2 mg/kg BW).

Initiation of phage therapy led to a rapid drop in CRP starting the day after the first dose. However, this was followed by an increase in inflammatory parameters together with an ongoing detection of VRE*fm* in the abdomen and respiratory tract. Associated therewith, quality control of the magistral phage preparation showed a rapid drop of phage titer with one log after 9 days (also see Section 3.2), indicating that the applied phage titers might have been lower than expected and did not reach levels throughout the treatment that were reported as being effective [24]. It was decided to prolong the treatment for another 10 days with a newly produced batch of magistral phage preparation with an increased phage concentration (joint titer 5.2 × 10^8^ PFU/mL in NaCl 0.9%) and administered 2 mL/kg BW twice daily intravenously. Following this, a persistent reduction in baseline CRP together with constant improvement of the clinical status was achieved. During and after treatment, we did not observe any adverse events attributable to phage administration. With the limitation that the intra-abdominal compartment was not accessible after permanent closure of the abdominal wall and removal of all drainages, for the rest of the hospital course, VRE*fm* was not detected in routine screening from rectal and tracheal (via tracheostoma) swabs.

After phage therapy, recovery of the patient was interrupted by a blood stream infection with *Klebsiella pneumoniae* (Figure 1, days 132–143) followed by a respiratory infection with *Enterobacter cloacae* (Figure 1, days 149–153). Both infections responded well to an escalation of the antibiotic treatment with meropenem. After five months of mechanical ventilation and unsuccessful attempts prior to phage therapy, the patient was successfully weaned from the respirator. Two weeks later, after a total of 167 days, she was discharged from the pediatric intensive care unit and was transferred to a rehabilitation center shortly after.

Long-term follow up was complicated by sustained intra-abdominal inflammation, which most likely was associated with ischemic-type biliary lesions after liver transplantation, which were present in sonography and represented by elevated gGT levels. This required a long-term oral therapy with empiric ciprofloxacin in combination with linezolid throughout the one year follow up. The latter antibiotic was added after routine screening had detected (one-time) the presence of VRE*fm* in rectal and tracheal swabs. Afterwards, VRE*fm* was not detected in any of the routine screenings, even after administration of linezolid was stopped, until submission of this work.

### 3.2. Phage Procurement and Quality Standards

Although phage therapy is increasingly identified as a promising tool for the treatment of MDR bacterial infections, the current lack of broadly available commercial (GMP-certified) phage preparations made us reach for a solution that consisted of an international academic collaboration, in order to obtain a personalized phage preparation with an acceptable quality.

An international call for suitable phages was coordinated by IPATH at UC San Diego, and bacterial isolates were sent to responding laboratories. Two lytic phages, EFgrKN (GenBank accession: MW004544.1) and EFgrNG (GenBank accession: MW004545.1) [17], were matched to the patient’s strain at the Israeli Phage Bank (IBP) at the Hebrew University of Jerusalem (Jerusalem, Israel).

Both phages delayed the growth of VRE*fm* by more than 12 h (Appendix A, and also as schematical figure in [18]). Additionally, sub-inhibitory concentration of the antibiotics gentamicin and chloramphenicol increased the inhibitory effect of both phages by 20 h and 4 h, respectively. The addition of sub-inhibitory concentration of vancomycin increased the inhibitory effect of phage EFGrKN, but not of EFGrNG, by an additional 10 h. Furthermore, in the case of EFGrKN with vancomycin and gentamycin, almost no regrowth was observed (Appendix A and also as schematical figure in [18]).

To minimize the risk of adding virulence factors in vivo, the absence of lysogeny was confirmed by BLAST analysis of the phage DNA sequence against known repressor or integrase sequences. In addition, the absence of sequences coding for known virulence factors, submitted to many databases, was tested using Abricate (version 0.8.13, Seemann T. Abricate Github https://github.com/tseemann/abricate (accessed on 30 July 2021)).

For the production of the phages as APIs, purified from the bacterial remnants (e.g., endotoxins) and fit for incorporation in magistral preparations for medical use (incl. intravenously) [20], the two phages were shipped to the Queen Astrid Military Hospital in Brussels (Brussels, Belgium). The produced phage APIs were diluted and aliquoted (magistral preparation) in vials with weight-adjusted dosages and were sent to the treating facility, the Department of Pediatrics at University Medical Center Hamburg Eppendorf (Hamburg, Germany), where they were stored at 4 °C until use.

The stability of the phage magistral preparations stored at 4 °C was evaluated over a 2-month period. The titer of the first batch (8.1 × 10^7^ PFU/mL) dropped with one log after 9 days and kept on decreasing, showing a more than 4 log reduction after 50 days. Thus, this preparation was probably not suitable for therapeutic application, as empirically an effective “therapeutic titer” for phage preparations is considered in the range of 10^6^–10^8^ PFU/mL [24]. The titer of the second batch (5.2 × 10^8^ PFU/mL) appeared to be more stable and dropped with one log PFU/mL after 63 days.

### 3.3. Humoral Immune Response

The response of the patient’s adaptive immune system towards the applied phages was tested using a classical phage neutralization assay [21] performed on serum samples collected 4, 28 and 49 days after initiation of phage therapy. Neutralizing phage antibodies were not detected (Figure 2). This implies that the phages EFgrKN and EFgrNG did not generate a strong immune response in the patient, particularly by stimulating synthesis of neutralizing antibodies. However, these results must be interpreted in the context of an overall immunosuppressed transplant patient with impaired capability of immunoglobulin synthesis.

### 3.4. Genetic Characterization of E. faecium Isolates

In vitro data of VRE*fm* indicated a synergistic effect of phage EFGrKN in combination with vancomycin (Appendix A). Importantly, over the course of the infection several additional VRE*fm* were isolated from various body sites (Figure 1, Table 1). Shortly after phage application, *E. faecium* exhibiting a vancomycin-susceptible phenotype was identified (Figure 1, Table 1). Vancomycin-resistant and susceptible isolates all belonged to ST1299, and core genome analysis only found an SNP distance of 4–12, suggesting that all isolates emerged from the same vancomycin-resistant progenitor. In vancomycin-susceptible *E. faecium*_1–3 and *E. faecium*_4, a deletion of four contigs comprising 20.8 kb with >99.9% identity to van-cluster carrying *E. faecium* plasmid sequences (e.g., accession number KX810025.1) was identified, reasonably explaining the observed vancomycin-susceptible phenotype.

These results provide unambiguous genetic evidence for persistence of the same invasive *E. faecium* clone throughout the infectious course, which also experienced a loss of vancomycin resistance. However, the temporal relationship of phage therapy and van-loss has to be interpreted with caution and might just be coincidental, as van loss is frequently observed in VRE infections, e.g., [25], and the first vancomycin-susceptible isolate was retrieved already on the same day of first phage application.

## 4. Discussion

Abdominal site infections after portoenterostomy and orthotropic liver transplantation are a common complication [3,4,5], and, especially, post-transplant cholangitis reduces graft survival significantly [6]. Colonization and subsequent infections with VRE in liver transplant patients are overall associated with high mortality, and current antibiotic-based therapeutic strategies to achieve decolonization have low success rates [26]. In animal models, phages were shown to have a high therapeutic efficacy [27,28,29] and significant tissue penetration [30,31]. Taking into account the long history of clinical use in Eastern Europe and recent case reports, phage therapy qualifies as a promising tool in the treatment of MDR bacterial infections. Unfortunately, to date, no commercial phage therapy products have made it to the market. However, with this case we illustrated that the currently established international networks are capable to quickly respond in certain selected cases. In order to meet increasing demands, there is an urgent need for globally commercialized phage products, GMP-certified and tested in randomized controlled trials, but also for an approval of pragmatic manufacturing processes for the production of personalized phage products in combination within a broad collaboration network.

## 5. Conclusions

After an internationally coordinated effort, phage therapy was applied in a critically ill pediatric patient that had undergone three successive liver transplants and had a persistent VRE*fm* infection. Over a one-year follow up, the treatment was not associated with any adverse events. Although the disease course was complex, clinical improvement was clearly linked to phage application. To our knowledge, we here present the first case of intravenous phage therapy against an infection with *E. faecium*. The clinical course and the data concerning immune response, phage stability and product safety provide evidence for the potential benefit arising from phage therapy. The recent interest in phage-therapy is driven by the growing number of refractory drug-resistant bacterial infections. However, there is a need for standardized data to analyze efficacy and safety of phage therapy as well as product quality. This would support decision making of health care providers and would help regulatory authorities to license phage products and adapt their regulatory framework to approve manufacturing processes of personalized (adapted) phage products, which would facilitate the use of phage therapy in a broader patient spectrum.

## Figures and Tables

**Figure 1 viruses-13-01785-f001:**
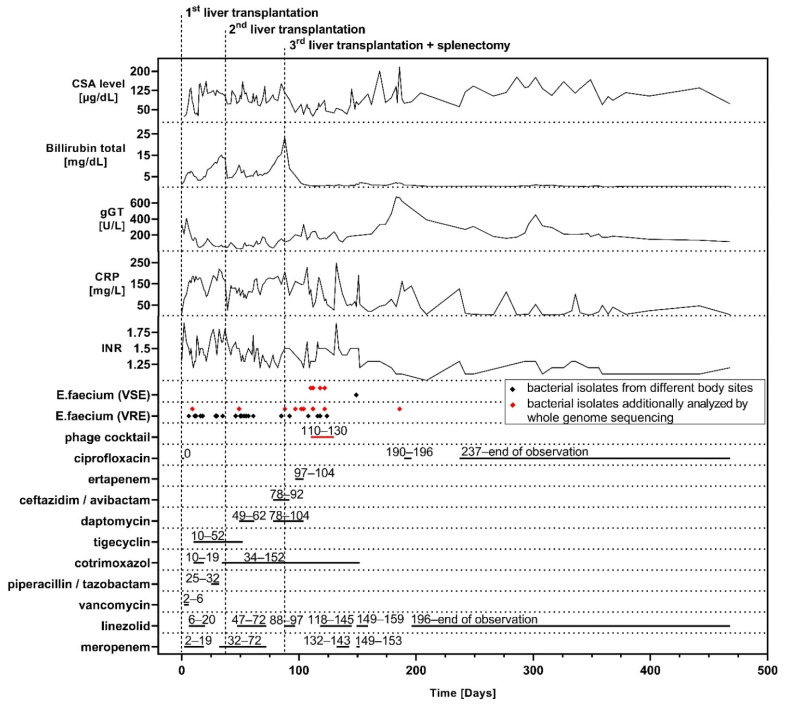
Time course of major operations, representative lab values, detection of vancomycin susceptible (VSE) and resistant (VRE) *E. faecium* from different sites as well as antibacterial treatment starting from the day of the first liver transplantation. Phages were administered over ten days of 2 mL/kg BW as a magistral preparation (joint titer first batch 8.1 × 10^7^ PFU/mL in NaCl 0.9%) followed directly by another ten days of 2 mL/kg BW (joint titer second batch 5.2 × 10^8^ PFU/mL in NaCl 0.9%) twice daily. Exact days of treatments, where the date of the first liver transplantation is defined as day 0, are plotted above each timeline, and isolates that were further analyzed by whole-genome sequencing are highlighted.

**Figure 2 viruses-13-01785-f002:**
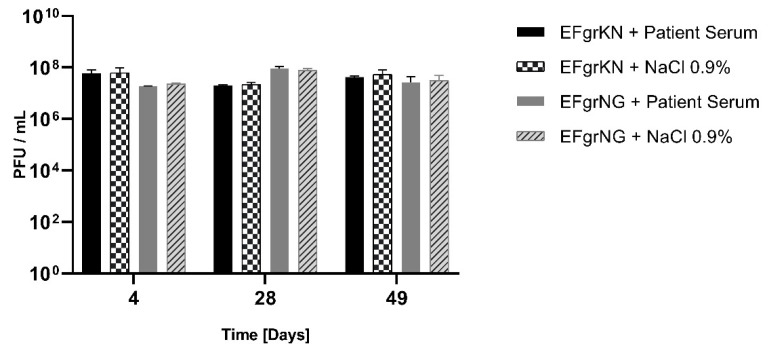
Phage neutralization assay was performed in triplicate on serum samples of the patient collected on days 4, 28 and 49 after initiation of phage therapy with the phages EFgrKN and EFgrNG. The resulting phage activity, after incubation with the serum samples, was expressed in PFU/mL and compared to a control consisting of phage particles in NaCl 0.9%, after incubation at 37 °C for 30 min. Results are plotted as mean ± SD. At no time point could relevant differences in phage titers, which would indicate neutralizing activity of patient serum, be observed.

**Table 1 viruses-13-01785-t001:** VRE*fm* and *E. faecium* isolates identified before and after phage therapy.

Isolate ID	Isolation Site	Isolation Relative to Phage Application (Days)	MIC ^1^	MLST ^2^
			Vancomycin (mg/L)	Sequence Type
VRE*fm*_1	Blood culture	−101	>256	1299
VRE*fm*_2	Central venous	−61	>256	1299
	catheter			
VRE*fm*_3	Abdominal swab	−21	>256	1299
VRE*fm*_4	Abdominal swab	−12	>256	1299
VRE*fm*_5	Abdominal drainage	−7	>256	1299
VRE*fm*_6	Upper respiratory tract swab	−7	>256	1299
VRE*fm*_7	Intraoperative swab	−5	>256	1299
*E. faecium*_1	Abdominal drainage	0	2	1299
*E. faecium* _2	Abdominal drainage	2	2	1299
*E. faecium*_3	Abdominal drainage	8	2	1299
*E. faecium* _4	Abdominal drainage	10	2	1299
VRE*fm*_8	Abdominal drainage	10	>256	1299
VRE*fm*_9	Inguinal swab	76	>256	1299

^1^ MIC—minimum inhibitory concentration. ^2^ MLST—multi-locus sequence typing.

## Data Availability

Phage Genome Sequences are accessible at GenBank; EFGrKN’s accession number is MW004544.1, and EFGrNG’s accession number is MW004545.1. Raw data are available over the NIH BioSample database project (BioProject number PRJNA706131)/Israeli Phage Bank (IPB), with accession numbers SAMN18137778 for EFGrKN and SAMN18191644 for EFGrNG. Further clinical data are available on request from the corresponding author. The data are not publicly available for protection of the patient’s identity.

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
