# Peer review of "Bacteriophage Rescue Therapy of a Vancomycin-Resistant Enterococcus faecium Infection in a One-Year-Old Child following a Third Liver Transplantation"

_viruses, 2021, doi:10.3390/v13091785_

Round 1

Reviewer 1 Report

The work is interesting due to the need of develop new strategies for  multidrug resistant bacterial infections.

However, I have some questions:

1) it is necessary to improve the introduction, material and methods, discussion and conclusions.

2) it is not describe in detail the treatment. Authors wrote "15 weeks after the first, and three weeks after the third liver transplantation, we were able to apply an individualized two-phage cocktail prepared for use as salvage therapy under the umbrella of article § 37 (Unproven Interventions in Clinical Practice) of the Declaration of Helsinki after expert advice and informed consent of the family".

- It is necessary to do an schematic representation of phage therapy including surgeries

- It is necessary to include information about the approval of the ethics committee

3) It is necessary to improve the explanation about the use of H1-antagonist

4) The results represented in Supplementary Figures 1,2 and 3.. are made them by only one individual assay? It is necessary at least a triplicate assay with statistical analysis.

5) In the Figure 2, should be necessary to include a positive control of the assay and add an statistical analysis.

Reviewer 2 Report

Dear authors,

I reviewed this interesting case report with favourable results after multiple surgical and non-surgical interventions, including administration of a 2 phage cocktail on top of antibiotics for treating recurrent vancomycin resistant infections produced by E.faecium. I would like to make the following minor comments for your consideration:

-line 60: I would not entirely agree that phage therapy was predominantly developed in Eastern Europe. See also e.g. here: https://www.ncbi.nlm.nih.gov/pmc/articles/PMC3442826/ I would suggest to e.g. replace the word developed with "used" or similar.

-line 61: it would be good to mention that the real recent interest in phage-therapy is probably driven by the growing number of refractory drug-resistant bacterial infections

-line 66: it should be clarified that the existing regulatory issues are not because regulators are not engaging in the process, but are linked to the fact that no convincing efficacy data are available (admittedly this is not the only issue, but is extremely important).

-Figure 1: you may wish to add the days of starting/stopping a particular therapy (e.g. D22 to D42) on top of the existing lines

-Line 200: looking at the text one would have expected meropenem to be given toward the end of therapy. This does not seem to be the case in figure 1; please clarify.

-Lines 276-291:  the discussion may need to mention that  while the favourable results in this particular case are welcome, compelling evidence of using phase therapy, in terms of quality, safety and efficacy (either as an alternative or as an adjunctive to antibacterial therapy) will be needed to be able to rely on this method and to expand its use to other patients. A mention could also be made in the conclusion.

-supplementary material: there is a typo in the title, please correct.

Round 2

Reviewer 1 Report

I agree with the responses